# Sleep Promoting Effects of Lettuce (*Lactuca sativa* L.) Extracts in Korean Adults with Poor Sleep Quality: A Randomized, Double-Blind Placebo-Controlled Trial

**DOI:** 10.3390/nu17132172

**Published:** 2025-06-30

**Authors:** Kumhee Son, Miji Lee, Min Kyung Bok, Kyoung Jin Hwang, Hyunjung Lim

**Affiliations:** 1Department of Medical Nutrition, Graduate School of East-West Medical Science, Kyung Hee University, Yongin 17104, Republic of Korea; sonkumhee@khu.ac.kr (K.S.);; 2Research Institute of Medical Nutrition, Kyung Hee University, Seoul 02447, Republic of Korea; 3Department of Neurology, Kyung Hee Medical Center, School of Medicine, Kyung Hee University, Seoul 02447, Republic of Korea

**Keywords:** sleep quality, Pittsburgh sleep quality index, sleep quantity, lettuce, heukharang

## Abstract

**Background/Objectives** Lettuce is known to contain compounds that promote sleep. This study aims to evaluate the effects of lettuce extract on Korean adults experiencing poor sleep quality. **Methods** In this randomized, double-blind, placebo-controlled trial, participants aged 30–65 with poor sleep quality (Pittsburgh Sleep Quality Index (PSQI) > 5) were recruited. Over 4 weeks, participants took two capsules daily of either the test extract or placebo. Sleep quality and quantity were assessed using the PSQI, actigraphy and polysomnography, and analyzed using ANCOVA adjusting for baseline, age, and sex. **Results** The adjusted final PSQI scores showed greater improvement in the test group than in the placebo group for both the global scores (6.48 ± 0.63 vs. 7.41 ± 0.57, *p* = 0.0462). Regarding actigraphy measurements, the adjusted final means showed significant improvements in the test group compared to the placebo group for total sleep time (TST) (421.68 ± 13.29 vs. 386.57 ± 12.27 min, *p* = 0.0023) and sleep efficiency (SE) (83.90 ± 1.6 vs. 81.01 ± 1.50%, *p* = 0.0342). Polysomnography results also favored the test group, with higher adjusted final means TST (358.90 ± 19.75 vs. 322.11 ± 17.66 min, *p* = 0.0457) and SE (86.86 ± 3.31 vs. 79.60 ± 2.99%, *p* = 0.0182), and lower wake after sleep onset (39.26 ± 10.57 vs. 68.15 ± 9.60 min, *p* = 0.0042). **Conclusions** Heukharang extract may enhance sleep quality and quantity and is deemed safe, suggesting its potential as a functional food for improving sleep.

## 1. Introduction

Insufficient sleep is a significant issue in modern society and poses a considerable public health challenge [1]. In South Korea, sleep problems are particularly severe, with Koreans averaging 41 min less sleep per day than individuals in other countries which are part of the Organization for Economic Cooperation and Development [2]. Furthermore, the prevalence of sleep disorders is increasing, with the number of affected individuals rising by 5.2% annually, from 543,184 in 2016 to 656,391 in 2020 [3]. This trend highlights the need for increased awareness and effective interventions to improve sleep health in Korea.

Sleep is a vital physiological process necessary for human health and well-being [4]. Sleep deprivation can lead to reduced immune function, neurodegenerative diseases, musculoskeletal diseases, diabetes, hypertension, and gastrointestinal issues, and it notably increases the incidence and mortality rates of cardiovascular diseases. Various factors, including the external environment and internal elements like the interaction between homeostasis and circadian rhythms, melatonin, and sleep-related biochemical substances, influence sleep disorders [5]. For chronic or severe sleep disorders, treatment typically involves medications and cognitive behavioral therapy [6]. Benzodiazepines (BDZ) are often prescribed, but long-term use can lead to side effects such as drug resistance, addiction, and depression [7]. In contrast, mild or short-term insomnia generally does not require medical treatment; instead, lifestyle and behavioral changes are advised, including a healthy diet, regular exercise, consistent sleep schedules, and reduced caffeine intake [8,9]. Recently, nonpharmacologic methods using natural plant extracts have been suggested for managing mild sleep disorders [9,10]. These extracts are being developed as functional health foods, offering an alternative treatment approach. However, their applications in adults with sleep disorders in Korea are still limited, indicating opportunities for further research and development in this area.

*Lactuca sativa* L. is a vegetable belonging to the Asteraceae family, and is a popular vegetable cultivated worldwide in various forms such as romaine lettuce, green leaf lettuce, and red leaf lettuce, and is often used in salads [11,12]. Lettuce not only has excellent nutritional value as it contains dietary fiber and various vitamins, but it is also known to contain hypnotic ingredients that may promote sleep [13]. The sleep-promoting effect is due to sesquiterpene lactones (BSLs), including lactucin and lactucopicrin, which give lettuce its bitter taste [14]. These ingredients act on γ-aminobutyric acid (GABA) receptors like the action of BDZ to promote physiological activities such as sedation and sleep promotion [15]. A new Korean lettuce variety called Heukharang has a lactucin concentration of 3.74 mg/g, which is 120 times higher than the 0.03 mg/g of regular lettuce [16]. Therefore, this variety has a stronger bitter taste, dark red leaf color, and anthocyanin color is evenly distributed throughout the leaf [16]. Lettuce showed a significant sedative effect and enhanced the hypnotic effect of barbiturates in animal models [17]. However, studies on its effects on human sleep are limited, and few clinical studies have measured its effects using both objective and subjective indicators [18,19]. In the preclinical trial of this study, the extract of Heukharang improved sleep latency and sleep duration through upregulation of GABA receptor subunits as well as adenosine A1 receptor in pentobarbital-injected mice [20]. This showed the potential of Heukharang to be developed as a functional food material. However, there has been no clinical study on the sleep-improving effect of Heukharang extract.

Therefore, this study aims to investigate the effects of lettuce extract derived from Heukharang as a functional food in Korean adults with poor sleep quality, using both subjective and objective sleep measures to comprehensively evaluate sleep improvement.

## 2. Materials and Methods

### 2.1. Study Design and Participants

This study was a randomized, double-blind, placebo-controlled trial to examine the effects of lettuce extract supplements on sleep quality over 4 weeks. Participants aged 30–65 with poor sleep quality (defined as a Pittsburgh Sleep Quality Index [PSQI] score > 5) [21] were recruited through announcements on hospital websites. Informed written consent was obtained from all participants before the study began. Individuals were excluded if they (1) had a high or low Insomnia Severity Index (ISI) [22] score (≥22 or ≤7); (2) were diagnosed with a sleep disorder or psychiatric condition; (3) had a body mass index (BMI) < 18.5 kg/m^2^ or ≥30.0 kg/m^2^; (4) underwent significant personal environmental changes that could cause severe mental stress; (5) had underlying conditions that could lead to sleep disorders; (6) traveled internationally with a time difference in the past month or planned to do so during the study; (7) worked shifts; (8) had uncontrolled hypertension or diabetes; (9) experienced renal or hepatic dysfunction; (10) regularly took medications, functional foods, or herbal treatments that could affect sleep; or (11) had excessive alcohol consumption (>140 g/week), smoked heavily (>10 cigarettes/day), or drank more than 10 cups of coffee daily.

The study took place from February 2021 to February 2022 at Kyung Hee University Hospital. The research protocol was approved by the Institutional Review Board of Kyung Hee University Hospital (IRB No: KHUH 2021-01-030-002, approved on 26 February 2021) and registered with the Clinical Research Information Service (CRIS; https://cris.nih.go.kr/cris/index/index.do, accessed on 11 May 2022), a primary registry in the WHO Registry Network, under the registration number KCT0007347 on 26 January 2021. The full protocol and statistical analysis plan can be accessed via the CRIS registry under this registration number.

Figure 1 presents the flow chart of the study. A total of 108 participants were screened, with 100 enrolling in the study. Participants were randomly assigned to either the test group (*n* = 50) or the placebo group (*n* = 50) during their first visit. Each participant visited the hospital four times: for the screening (2 weeks) and then at 0, 3, and 4 weeks. During the study, three participants in the test group dropped out due to low compliance and inadequate actigraphy wear. In the placebo group, six participants withdrew due to low compliance, withdrawal of consent, and loss of follow-up. Consequently, the data analysis included a total of 91 participants.

### 2.2. Sample Size

The sample size was calculated based on the primary outcome, the change in PSQI score between the test and placebo groups after 4 weeks. According to a previous study using a similar functional product [23], the expected mean (± SD) PSQI scores were 10.0 ± 2.1 in the test group and 11.7 ± 1.8 in the placebo group. Using G*Power software (version 3.1.9.2), a total sample size of 72 participants was required to detect a significant difference between groups with an effect size of 0.87, at a two-sided significance level of 0.05 and a power of 95%. Considering a dropout rate of approximately 28%, we planned to recruit 100 participants (50 per group).

### 2.3. Randomization

Participants who passed the screening process were randomly assigned to either the test or placebo group using a stratified block randomization method with a 1:1 allocation ratio. Stratification was applied based on participants’ willingness to undergo PSG: those who opted for PSG were allocated under IDs R001–R040, and those who declined PSG were allocated under IDs R041–R100. Randomization sequences were generated by an independent statistician and concealed in a sealed allocation list until the day of intervention (Day 1). Study personnel involved in enrollment and outcome assessment were blinded to group allocation. Intervention products were identical in appearance and labeled with participant codes according to the allocation list to maintain blinding.

### 2.4. Test Product

The test products used in this study were prepared by extracting lettuce (*Lactuca sativa* L.) with 30% ethanol for 4 h, followed by concentration to 12 Brix and subsequent freeze-drying. Each capsule of the test product contained 500 mg (50%) of the lettuce extract, 345 mg (34.5%) of microcrystalline cellulose, and 100 mg (10%) of maltodextrin. The placebo product was composed of 545 mg (54.5%) of microcrystalline cellulose and 400 mg (40%) of maltodextrin. The test products were visually and organoleptically indistinguishable from the placebo in terms of color, shape, and taste. Participants were instructed to take two tablets of either the test or placebo product 30 min before bedtime daily over a period of 4 weeks.

Participants, study investigators, outcome assessors, and data analysts were all blinded to group assignments. The test and placebo products were identical in packaging and labeling, and randomization codes were held by an independent coordinator until the end of the study to ensure blinding.

### 2.5. Sleep Quality Assessment

Sleep quality was evaluated using the PSQI, a validated self-report questionnaire designed to evaluate sleep quality, consisting of 19 items divided into seven components: sleep quality, sleep latency, sleep disturbances, sleep duration, sleep efficiency, use of sleeping medication, and daytime dysfunction. Each component is scored from 0 to 3, with a total score ranging from 0 to 21. A PSQI score greater than 5 indicates poor sleep quality.

Additionally, the ISI, a self-report tool for assessing insomnia [22]; the Epworth Sleepiness Scale (ESS), a self-report tool for measuring daytime sleepiness [24]; and Stanford Sleepiness Scale (SSS), a self-report tool for evaluating overall alertness throughout the day [25], were used to assess sleep quality. All assessments were conducted at baseline and at the end of the study, with higher scores reflecting negative outcomes.

### 2.6. Sleep Quantity Assessment

Sleep quantity was evaluated using actigraphy (Actiwatch Spectrum Plus, Philips Electronics Ltd., Hamburg, Germany), a sleep diary, and polysomnography (PSG) to assess the differences between groups in total sleep time (TST, min), sleep efficiency (SE, %), and wake after sleep onset (WASO, min). All participants wore watch-type actigraphy to monitor sleep quantity for 1 week prior to the study and 1 week before its conclusion. Participants recorded their sleep diary during the same period as the actigraphy.

PSG is a reliable and valid objective assessment tool that provides a comprehensive evaluation of sleep patterns [26]. However, it was not feasible to apply PSG to all participants due to cost and space limitations. Thus, 40 volunteers willing to undergo PSG were randomly assigned to the test group (*n* = 20) or the placebo group (*n* = 20), with PSG conducted at baseline and at the end of the study. Participants were hospitalized at Kyung Hee University Hospital for overnight PSG testing and discharged the following day. Caffein and alcohol consumption were prohibited for 1 week before the test.

### 2.7. Assessment of Sleep-Related Symptoms

Sleep-related symptoms were assessed using the Korean version of the Beck Depression Inventory (BDI) to evaluate the severity of depressive symptoms [27], the Beck Anxiety Inventory (BAI) to measure anxiety severity [28], and the Korean Fatigue Severity Scale (FSS) to assess the impact of fatigue on participants [29]. All three questionnaires were conducted at both baseline and at the end of the study. In all three assessments, higher scores indicate worse outcomes.

### 2.8. Dietary Intake and Physical Activity

To assess daily dietary intake, participants were asked to complete a 3-day food diary at baseline and at the end of the study, recording the name of each food item, its ingredients, and the amount consumed on two weekdays and one weekend day. Caffeine and alcohol intake, if any, were also recorded, as these substances could potentially affect the study outcomes. Daily nutrient intake was analyzed using a computer-aided nutritional analysis program (CAN-Pro 5.0; Korean Nutrition Society, Seoul, Republic of Korea).

Physical activity was also assessed at baseline and at the end of the study using the International Physical Activity Questionnaire (IPAQ) [30].

Throughout the study, participants were advised to maintain their usual levels of physical activity and dietary habits, including caffeine and alcohol intake, except for the 1 week prior to the sleep test, during which both were restricted.

### 2.9. Compliance Assessment

Compliance with the study protocol was evaluated based on the amount of test or placebo product remaining at the end of the study. Compliance rate less than 70% were classified as “low”.

### 2.10. Safety Assessment

Safety assessments for both the test and placebo products were conducted at multiple time points. Biochemical tests were conducted at screening (within 14 days prior to study initiation) and on the final day of the study. Blood samples were drawn from the mid-arm vein after a 12 h overnight fast for safety evaluation. Samples were collected in ethylenediaminetetraacetic acid-potassium anticoagulant tubes and serum separator tubes (SST). After allowing the blood to clot for 30 min, SST samples were centrifuged at 3000× *g* at 4 °C for 10 min, and the supernatant was analyzed. AST and ALT levels were measured using UV without P5P (Qualigent AST-L and Qualigent ALT-L, Sekisui, Tokyo, Japan).

Participants’ clinical conditions and adverse events (AEs) were monitored throughout the study period through medical interviews conducted by physicians and self-reports provided by the participants.

Blood pressure and pulse rate were recorded twice during each visit using an automated blood pressure monitor (HEM-7156, Omron, Hà Nội, Vietnam) while the participant was seated and at rest for over 15 min. The average of the two readings was used for analysis.

### 2.11. Statistical Analysis

The results were analyzed using the per-protocol (PP) set for efficacy evaluation with SAS^®^ (Version 9.4, SAS Institute, Cary, NC, USA). Continuous variables related to general characteristics, such as age, PSQI score, BMI, dietary intake, and physical activity, were presented as mean ± standard deviation (SD) and compared between groups using Student’s *t*-test. Categorical variables were expressed as numbers and percentages and analyzed using the Chi-square test or Fisher’s exact test to assess the differences in sex distribution and health-related behaviors (e.g., drinking, smoking) between the test and placebo groups.

The primary evaluation of the test product’s efficacy was conducted using linear regression models, adjusting for baseline values, age, and sex. Results were presented as adjusted means with standard errors (SE), and between-group differences were expressed using 95% confidence intervals (CI). Additionally, within-group changes before and after the intervention were assessed using paired *t*-tests or Wilcoxon signed-rank tests. A *p*-value of <0.05 was considered statistically significant.

## 3. Results

### 3.1. General Characteristics Measurement

There were no significant differences in baseline characteristics between the test and placebo groups, including age, sex, anthropometric measurements, dietary intake, physical activity, alcohol consumption, smoking status, and caffeine intake (Table 1). These variables were effectively controlled during the study; there were no significant differences between the groups at the end of the study.

### 3.2. Sleep Quality

The results for the PSQI total score and its components are presented in Table 2. The adjusted final mean PSQI scores were significantly lower in the test group (6.48 ± 0.63) than in the placebo group (7.41 ± 0.58), indicating greater improvement in sleep quality. Table 3 presents the outcomes for ISI, ESS, and SSS. No significant differences in adjusted final mean scores were observed between the groups. Within-group comparisons revealed significant improvement in all three scores in the test group, whereas the placebo group showed significant changes in ISI and ESS scores only, with no significant changes in SSS (Appendix A). These findings suggest a more pronounced enhancement in alertness in the test group compared to the placebo group.

### 3.3. Sleep Quantity

The TST, SE, and WASO measured by actigraphy and PSG are summarized in Table 4 and illustrated in Figure 2. Based on actigraphy data, the adjusted final mean TST was longer in the test group than in the placebo group (421.68 ± 13.29 vs. 386.57 ± 12.27 min, *p* = 0.0023), and SE was also higher (83.90 ± 1.60 vs. 81.01 ± 1.50%, *p* = 0.0342). No significant difference in WASO was found between the two groups.

According to PSG measurements, the test group showed longer TST (358.90 ± 19.75 vs. 322.11 ± 17.66 min, *p* = 0.0457), higher SE (86.86 ± 3.31 vs. 79.60 ± 2.99%, *p* = 0.0182), and shorter WASO (39.26 ± 10.57 vs. 68.15 ± 9.60 min, *p* = 0.0042) compared to the placebo group.

These findings suggest that the test product led to greater improvements in objective sleep quantity measures.

### 3.4. Sleep-Related Symptoms

Table 5 shows that there were no statistically significant differences in the adjusted final mean scores of BDI, BAI, and FSS between the test and placebo groups. Within-group comparisons revealed significant improvement in all three scores in the test group, whereas the placebo group showed significant improvements in BDI and BAI scores only, with no significant changes in FSS (Appendix A). These findings suggest that while both groups experienced reductions in depression and anxiety, only the test group demonstrated a significant improvement in fatigue.

### 3.5. Compliance

Compliance rates were 95.52 ± 4.43% in the test group and 92.56 ± 6.74% in the placebo group. However, two participants (one from each group) with compliance rates below 70% were excluded from the final analysis.

### 3.6. Safety Outcomes

The safety assessment indicated that participants’ vital signs and blood biochemical profiles remained within normal limits before and after the study (Appendix A). No severe AEs were reported during the study period, although one minor AE of dyslipidemia was noted in the placebo group and determined to be unrelated to the study.

## 4. Discussion

This randomized controlled trial demonstrated that Heukharang (*Lactuca sativa* L.) extract, administered for 4 weeks, significantly improved both sleep quality and quantity in Korean adults with poor sleep quality. The adjusted final PSQI scores, a key indicator of subjective sleep quality, were significantly lower in the test group compared to the placebo group. Similarly, objective measures of sleep quantity—TST, SE, and WASO—showed significantly greater improvements in the test group. Additionally, significant improvements in daytime alertness and fatigue symptoms were observed only in the test group from baseline to the end of the study.

Lettuce (*Lactuca sativa* L.), known to contain bioactive compounds such as lactucin and lactucopicrin, has been suggested to exert sedative and anxiolytic effects by modulating GABA levels and its receptor activity in the brain [31]. In particular, the GABAA receptor, a fast-acting inhibitory ionotropic receptor, plays a critical role in the action of benzodiazepines, barbiturates, and steroids in the mammalian central nervous system [32]. Previous in vitro studies demonstrated that lactucin and lactucopicrin exhibit high binding affinities to the GABAA-BDZ receptor (80.7 ± 2.3% and 55.9 ± 0.7%, respectively), contributing to prolonged total sleeping time [17]. Several clinical studies have also explored the sleep-promoting effects of lettuce extract in human populations [18,19]. Mosavat et al. conducted a 4-week intervention using lettuce syrup (5 mL twice a day) in breast cancer patients, reporting a significant reduction in PSQI scores in the test group (from 12.76 ± 2.31 to 10.24 ± 4.12, *p* = 0.014), while changes in the control group were not significant (12.27 ± 2.05 to 10.11 ± 4.49, *p* = 0.076) [19]. Similarly, Pour et al. reported improved subjective sleep quality in pregnant women with insomnia after a 2–week administration of lettuce extract capsules, without adverse effects [18]. Although both studies demonstrated improvements in sleep quality, they relied solely on subjective assessments such as the PSQI, limiting comparability with the present study. In contrast, the current study is the first to assess both subjective (PSQI) and objective (actigraphy, PSG) outcomes to comprehensively evaluate the sleep-enhancing potential of lettuce extract. PSG, the gold standard for sleep evaluation, provides detailed physiological data including TST, SE, and WASO [33], while actigraphy offers a reliable field-based measure of sleep–wake patterns over time [34]. The incorporation of these objective tools, alongside validated subjective indices, enhances the robustness of our findings and highlights the need for similarly well-designed clinical trials. Moreover, the observed improvements in daytime alertness and reduced fatigue symptoms in the test group suggest a broader benefit of Heukharang extract beyond nocturnal sleep parameters. When evaluating sleep-promoting effects, it is important to consider not only hypnotic effect but also the ability to alleviate factors that interfere with sleep [35]. Finally, the efficacy of Heukharang may stem from its high concentration of bioactive sesquiterpene lactones (BSLs)—reportedly 100 times higher than standard lettuce—and its additional affinity for adenosine A1 receptors, which may further enhance sleep regulation via non-GABAergic pathways [36].

Many studies have also been conducted for improvement of sleep through GABAergic mechanisms [34,35,36,37,38,39,40]. Kim et al. evaluate the efficacy of alpha-s1 casein hydrolysate (ACH), a milk-derived peptide, in 48 Korean adults with poor sleep quality [37]. After 4 weeks of supplementation, significant improvements were observed in the test group compared to the placebo group in sleep diary-derived outcomes such as TST (*p* < 0.001), SE (*p* <0.001), and SL (*p* = 0.001). However, no significant differences were found in PSQI scores, insomnia severity, or psychiatric assessments between the two groups [37]. Moreover, although actigraphy showed an increase in SE (*p* = 0.031), PSG did not indicate significant intergroup differences [37]. These findings suggest inconsistencies between subjective and objective sleep indicators, with benefits limited to select parameters. Similarly, Um et al. examined the effects of rice bran oil extract in 50 adults with sleep disturbances and reported significant improvements in PSG-derived TST (*p* = 0.010), SE (*p* = 0.019), and SL (*p* = 0.047), although no corresponding improvements were observed in subjective outcomes [39]. Both ACH and rice bran oil extract are known to be able to enhance GABAergic neurotransmission through increased binding affinity to GABA_A receptors, yet the overall pattern of improvements was less comprehensive compared to those observed in the current study [37,39]. This disparity suggests that sleep-promoting efficacy may vary across GABAergic compounds and further highlights the potential superiority of Heukharang extract in modulating both subjective and objective sleep parameters. Additionally, Suanjiaorentang, a traditional herbal formula with purported GABAergic properties, was reported to be more effective than BDZ in improving insomnia symptoms. However, a systematic review noted that the quality of the supporting randomized controlled trials was methodologically insufficient to draw definitive conclusion [38].

Several studies have explored the sleep-promoting effects of functional substances acting through alternative biological pathways [41,42]. Eckert et al. investigated the effect of fish hydrolysate supplementation on sleep quality in healthy adults [41]. Fish hydrolysate is known to exert anxiolytic-like effects by limiting stress, induced increases in plasma corticosterone, inhibiting mitochondrial alterations, and modulating hippocampal stress response gene expression. After a 4-week intervention, subjective sleep quality improved significantly in the test group compared to the placebo group. However, objective indicators such as sleep duration, sleep phases, and sleep score measured using Fitbit did not show significant changes [41]. The lack of objective improvements may be attributed to an overestimation of the expected effect size during sample size calculation, as well as potential influences from external variables such as seasonal variation and the COVID-19 pandemic [41]. Murakami et al. evaluated the sleep-promoting effects of *Bifidobacterium adolescentis* SBT2786 in 126 stressed Japanese adults [42]. After a 4-week of supplementation, the test group exhibited a significant increase in objective measured sleep duration compared to the placebo group (*p* < 0.05). However, improvements in subjective sleep quality were only observed in a high-stress subgroup (*n* = 26), limiting the generalizability of the findings [42]. While these studies suggest that non-GABAergic functional components may support sleep regulation, their effects appear to be context-dependent and constrained by methodological limitations such as small sample sizes, narrow target populations, and short intervention periods.

This study has several notable strengths. It was the first double-blind, randomized, placebo-controlled clinical trial to evaluate the sleep-promoting effects of Heukharang lettuce extract. Unlike prior studies, this trial incorporated both subjective and objective sleep parameters, and improvements were consistently observed across validated outcomes. Subjective assessments such as the PSQI, and objective measures including TST, WASO, and SE using actigraphy and PSG, demonstrated that the extract improved both sleep quality and quantity. Furthermore, the observed reductions in fatigue-related symptoms support the broader benefits of the intervention on daily functioning. The safety profile was also acceptable, with no adverse events related to the intervention reported.

Although this study has notable strengths, some limitations remain. First, while preclinical evidence suggests that lettuce extracts may act through GABAergic pathways, direct assessment of such mechanisms was not feasible in this trial due to the lack of validated biomarkers for GABAergic activity in human subjects. However, the use of validated tools such as the PSQI and the gold-standard PSG strengthens the reliability of the findings and partially compensates for this limitation. Second, although PSG was administered to only a subset of participants due to cost and space limitations, all PSG-derived indicators showed significant improvements in the test group. Moreover, the actigraphy-based TST and SE outcomes were consistent with those obtained via PSG, reinforcing the credibility of the results despite partial discordance in WASO values. Lastly, most participants were female, which may limit the generalizability of the findings and necessitates replication in more balanced populations.

## 5. Conclusions

In conclusion, this randomized, double-blind, placebo-controlled trial suggests that Heukharang lettuce extract may have beneficial effects on sleep quality in adults with poor sleep. Further studies in populations with more balanced sex distribution are needed to validate and generalize these findings.

## Figures and Tables

**Figure 1 nutrients-17-02172-f001:**
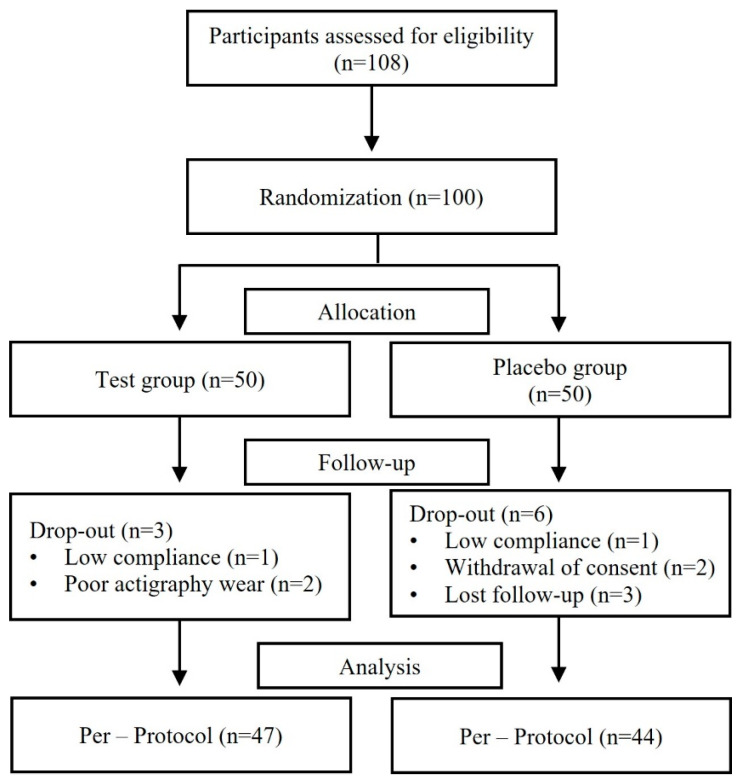
CONSORT flow diagram.

**Figure 2 nutrients-17-02172-f002:**
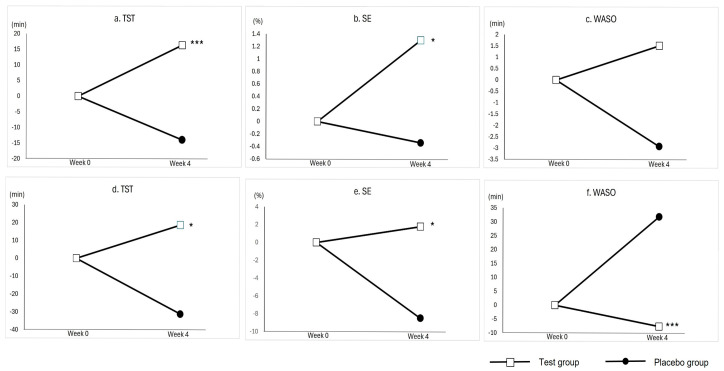
Adjusted mean changes in objective sleep quantity measures from baseline to week 4. (**a**) Total sleep time measured by actigraphy. (**b**) Sleep efficiency measured by actigraphy. (**c**) Wake after sleep onset measured by actigraphy. (**d**) Total sleep time measured by polysomnography. (**e**) Sleep efficiency measured by polysomnography. (**f**) Wake after sleep onset measured by polysomnography. *p*-values indicate between-group differences based on ANCOVA: * *p* < 0.05, *** *p* < 0.001.

**Table 1 nutrients-17-02172-t001:** General characteristics of the participants at baseline.

	Test Group (*n* = 47)	Placebo Group (*n* = 44)
Sex (*n* (%))		
Female	45 (96)	40 (91)
Age (years)	52.0 ± 8.0	53.9 ± 6.8
PSQI score	12.4± 2.0	11.8 ± 2.4
BMI (kg/m^2^)	23.1 ± 2.6	24.1 ± 3.1
Dietary intake		
Energy (kcal)	1678.0 ± 375.4	1628.5 ± 447.1
Carbohydrate (g)	223.0 ± 49.6	222.1 ± 76.2
Fat (g)	53.4 ± 18.2	47.0 ± 17.7
Protein (g)	67.7 ± 20.9	65.3 ± 16.4
Physical activity (MET-min/week)	1808.4 ± 1134.4	2254.8 ± 1661.0
Alcohol consumption (*n* (%))		
Yes	16 (34)	20 (44)
No	31 (66)	24 (56)
Smoking (*n* (%))		
Yes	1 (2)	3 (7)
No	46 (98)	41 (93)
Caffeine (cup/day)	1.3 ± 1.1	1.7 ± 1.3

PSQI, Pittsburgh Sleep Quality Index; BMI, body mass index. MET-min/week = metabolic equivalent minutes per week, calculated using IPAQ scoring protocol. Values are presented as means ± SD or *n* (%). No significant between-group differences were found in any of the variables based on Student’s *t*-test.

**Table 2 nutrients-17-02172-t002:** Comparison of PSQI scores between groups after adjustment for baseline, sex, and age.

	Test Group (*n* = 47)	Placebo Group (*n* = 44)	Group Difference	*p*-Value
Subjective sleep quality	1.23 ± 0.12	1.17 ± 0.11	0.06 (−0.15, 0.27)	0.5690
Sleep latency	1.12 ± 0.24	0.45 ± 0.22	−0.33 (−0.73, 0.06)	0.0991
Sleep duration	1.42 ± 0.19	1.68 ± 0.17	−0.27 (−0.59, 0.05)	0.0949
Habitual sleep efficiency	0.46 ± 0.23	0.71 ± 0.71	−0.25 (−0.64, 0.14)	0.2046
Habitual sleep efficiency (%)	0.89 ± 0.03	0.85 ± 0.03	0.05 (−0.00, 0.09)	0.0577
Sleep disturbances	1.17 ± 0.09	1.12 ± 0.08	0.05 (−0.11, 0.20)	0.5368
Use of sleep medication	–	–	–	–
Daytime dysfunction	1.02 ± 0.18	1.25 ± 0.17	−0.23 (−0.54, 0.08)	0.1405
Global score (range 0–21)	6.48 ± 0.63	7.41 ± 0.57	−0.93 (−1.99, −0.00	0.0462

PSQI, Pittsburgh Sleep Quality Index. Values are presented as adjusted means ± standard errors (SE) for each group, and as adjusted group differences with 95% confidence intervals (CI), all estimated using analysis of covariance, adjusting for baseline values, sex, and age. *p*-values indicate between-group differences based on ANCOVA.

**Table 3 nutrients-17-02172-t003:** Comparison of sleep quality (ISI, ESS, SSS) between groups after adjustment for baseline, sex, and age.

	Test Group (*n* = 47)	Placebo Group (*n* = 44)	Group Difference	*p*-Value
ISI	6.40 ± 0.86	6.42 ± 0.79	−0.03 (−1.47, 1.42)	0.9679
ESS	4.08 ± 0.61	4.58 ± 0.57	−0.50 (−1.53, 0.54)	0.3409
SSS	1.94 ± 0.21	2.21 ± 0.19	−0.27 (−0.63, 0.08)	0.1279

ISI, Insomnia severity index; ESS, Epworth sleepiness scale; SSS, Stanford sleepiness scale. Values are presented as adjusted means ± standard errors (SE) for each group, and as adjusted group differences with 95% confidence intervals (CI), all estimated using analysis of covariance, adjusting for baseline values, sex, and age. *p*-values indicate between-group differences based on ANCOVA.

**Table 4 nutrients-17-02172-t004:** Comparison of sleep quantity between groups after adjustment for baseline, sex, and age.

	Test Group (*n* = 47)	Placebo Group (*n* = 44)	Group Difference	*p*-Value
Actigraphy				
TST (min)	421.68 ± 13.29	386.57 ± 12.27	35.11 (12.92, 57.29)	0.0023
SE (%)	83.90 ± 1.60	81.01 ± 1.50	2.89 (0.2208, 5.56)	0.0342
WASO	50.43 ± 4.72	46.83 ± 4.34	3.60 (−4.23, 11.42)	0.3634
Sleep diary				
TST (min)	396.24 ± 8.11	379.54 ± 7.47	16.70 (3.15, 30.24)	0.0163
Sleep on latency	26.38 ± 4.04	27.08 ± 3.70	−0.70 (−7.48, 6.08)	0.8383
Number of awakenings	1.06 ± 0.14	1.09 ± 0.12	−0.03 (−0.26, 0.19)	0.7754
Subjective sleep quality	2.83 ± 0.12	2.93 ± 0.12	−0.10 (−0.31, 0.11)	0.3426
PSG				
TST (min)	358.90 ± 19.75	322.11 ± 17.66	36.79 (0.75, 72.83)	0.0457
SE (%)	86.86 ± 3.31	79.60 ± 2.99	7.26 (1.31, 13.20)	0.0182
WASO	39.26 ± 10.57	68.15 ± 9.60	−28.90 (−47.99, −9.80)	0.0042

TST, Total sleep time; SE, Sleep efficiency; WASO, Wake after sleep onset; PSG, Polysomnography. Values are presented as adjusted means ± standard errors (SE) for each group, and as adjusted group differences with 95% confidence intervals (CI), all estimated using analysis of covariance, adjusting for baseline values, sex, and age. *p*-values indicate between-group differences based on ANCOVA.

**Table 5 nutrients-17-02172-t005:** Comparison of sleep-related symptoms between groups after adjustment for baseline, sex, and age.

	Test Group (*n* = 47)	Placebo Group (*n* = 44)	Group Difference	*p*-Value
BDI	8.60 ± 1.23	9.99 ± 1.14	−1.33 (−3.40, 0.74)	0.2056
BAI	5.13 ± 0.97	5.73 ± 0.89	−0.60 (−2.23, 1.03)	0.4664
FSS	22.07 ± 1.95	24.13 ± 1.82	−2.05 (−5.34, 1.23)	0.2174

BDI, Beck Depression Inventory; BAI, Beck Anxiety Inventory; FSS, Fatigue Severity Scale. Values are presented as adjusted means ± standard errors (SE) for each group, and as adjusted group differences with 95% confidence intervals (CI), all estimated using analysis of covariance, adjusting for baseline values, sex, and age. *p*-values indicate between-group differences based on ANCOVA.

## Data Availability

The data presented in this study are openly available as Appendix A.

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
