# Peer review of "Sleep Promoting Effects of Lettuce (Lactuca sativa L.) Extracts in Korean Adults with Poor Sleep Quality: A Randomized, Double-Blind Placebo-Controlled Trial"

_nutrients, 2025, doi:10.3390/nu17132172_

Round 1
Reviewer 1 Report
Comments and Suggestions for Authors
This study was randomized, double-blind, placebo-controlled study testing the impact of a lettuce extract on sleep quality in adults with poor sleep. The study design is strong but information on some methods and more robust statistical analyses are warranted. Some findings appear to be regression to the mean and this should be verified.
Abstract
Please provide data on change in each group for SSS as done for other outcomes.
Methods
Why was a parallel-arm design chosen instead of a more robust crossover, within-subjects, design?
Lines 89, 95: Please remove/verify/explain the text “(Death, #4)”.
Section 2.5: Please mention when these questionnaires were administered during the intervention.
Section 2.6: Please be more precise in describing the time of safety assessments (e.g. how many days before and after the study?).
Please describe how dietary intakes and physical exercise were measured.
A stronger statistical approach would be to evaluate endpoint outcomes, adjusting for baseline values and covariates (age, sex) in linear regression models.
Results
Section 3.4: Please report on group differences. In this case, there was no difference between groups in BDI and BAI while there was a trend, not significant, for greater improvement in fatigue in the test group.
Were participants queried to determine if they were able to discern which group they had been assigned to?
Discussion
Please expand on the limitations of this study. The sample consisted of mostly women; the actigraphy and PSG data were not all concordant (e.g. SE); and no mechanisms were assessed. Please also include information on the strengths beyond design (double-blind and placebo-controlled).
Replication of this study is necessary and conclusions should be tempered. Many of the findings seem to reflect a regression to the mean (endpoint values are not different between groups).
Comments on the Quality of English LanguageThe discussion needs to be organized in paragraphs. One paragraph spans multiple pages.
Reviewer 2 Report
Comments and Suggestions for Authors
This is an interesting study regarding the use of Lactuca sativa L. extracts to improve sleep in Korean adults using polysomnography as an assessment tool. My comments on improving the manuscript are as follows:
Title: I recommend shortening the title. For example, you could trim it to: Sleep promoting effects of lettuce (Lactuca sativa L.) extracts in Korean adults: a randomized, double-blind placebo- controlled trial
Abstract: I suggest mentioning that the participants were Korean adults. In line 27, did you mean to say “deemed safe" rather than “deemed sage"?
Introduction: The introduction explains the background well and provides support for the study significance. In line 72, I recommend changing “In the preclinical study of this study” to “In the preclinical trial of this study” to avoid the repetition of the word “study”.
Methods: In line 89 and 95, “Death #4” sounds out of place. I assume this was a mistake. Throughout the methods, it would sound better to replace the terms “test group” and “test product” with “experimental group” and “experimental product”. The extraction protocol for the bioactive chemicals is described under section 2.2 Intervention and Compliance. I suggest creating a separate section for the extraction and compounding methods. In line 115 you mentioned that you are “extracting lettuce” but really you are extracting the bioactive compounds from the lettuce. Rather than call the bioactive compounds “lettuce extract”, it would be better to be more specific and call it lactucin as described in the introduction. Did you perform any analytical techniques (chromatography or spectroscopy) to verify the identity of the compounds extracted from the lettuce, such as HPLC or nuclear magnetic resonance? Were any standards used to confirm the identity? In lines 122-123, it says “Throughout the study, participants were advised to maintain their usual levels of physical activity and dietary habits, including their caffeine and alcohol intake.” However, in line 154 it says “Caffein and alcohol consumption were prohibited for 1 week before the test.” In the previous statement (line 122) you can add “except for 1 week before the X test” to be consistent.
Results: In table 2, one of the footnotes states “Significant differences at final between the groups by Student’s t-test at § p < 0.05, §§ p < 0.01, §§§ p < 0.001.” I don’t see these § symbols anywhere in the table. The results marked with the † symbol were hard to find within the table. I suggest making these a superscript within the table (like you did with the asterisks) or marking them in bold or another color such as red to make them more noticeable. In table 3, you have the footnote “Significant differences at baseline between the groups by Student’s t-test at † p < 0.05, †† p < 0.01, ††† p < 0.001” but I do not see any of these symbols in the table. Similar for Table 5, there are footnotes with symbols that are not actually used in the table, so this is just taking up space.
In the discussion line 370, you mention “It can be assumed that this study demonstrates the sleep-improving effects through upregulation of Adenosine A1 receptors and GABAA receptors, which were shown in preclinical study”. You may be able to suggest this as a possible mechanism, but should not assume it is occurring in humans based only on the preclinical study.
